# Development and Validation of a Prognostic Model for Lung Adenocarcinoma Based on CAF-Related Genes: Unveiling the Role of COX6A1 in Cancer Progression and CAF Infiltration

**DOI:** 10.3390/ijms26083478

**Published:** 2025-04-08

**Authors:** Xinyu Zhu, Bo Li, Lexin Qin, Tingting Liang, Wentao Hu, Jianxiang Li, Jin Wang

**Affiliations:** School of Public Health, Suzhou Medicine College of Soochow University, Suzhou 215123, China; 2330506091@stu.suda.edu.cn (X.Z.); 2330412052@stu.suda.edu.cn (B.L.); 2330506055@stu.suda.edu.cn (L.Q.); 2230412060@stu.suda.edu.cn (T.L.); 2330506024@stu.suda.edu.cn (W.H.)

**Keywords:** tumor-associated fibroblasts, immune microenvironment, lung adenocarcinoma, prognostic model, COX6A1

## Abstract

Lung adenocarcinoma (LUAD), the predominant subtype of non-small cell lung cancer (NSCLC), presents significant challenges in early diagnosis and personalized treatment. Recent research has focused on the role of the tumor microenvironment, particularly tumor-associated fibroblasts (CAFs), in tumor progression. This study systematically analyzed CAF immune infiltration-related genes to construct a prognostic model for LUAD, confirming its predictive value for patient outcomes. The risk score derived from CAF-related genes (CAFRGs) was negatively correlated with immune microenvironment scores and linked to the expression of immune checkpoint genes, indicating that high-risk patients may exhibit immune escape characteristics. Analysis via the TIDE tool revealed that low-risk patients had more active T-cell immune responses. The risk score also correlated with anti-tumor drug sensitivity, particularly to doramapimod. Notably, *COX6A1* emerged as a key gene in the model, with its upregulation associated with immune cell infiltration and immune escape. Further in vitro experiments demonstrated that *COX6A1* regulates LUAD cell migration, proliferation, and senescence, suggesting its role in tumor immune evasion. Additionally, further co-culture studies of lung cancer cells and fibroblasts revealed that *COX6A1* knockdown promotes the expression of CAF-related cytokines, enhancing CAF infiltration. Overall, this study provides a foundation for personalized treatment of LUAD and highlights *COX6A1* as a promising therapeutic target within the tumor immune microenvironment, guiding future clinical research.

## 1. Introduction

Lung adenocarcinoma (LUAD) is the most common subtype of non-small cell lung cancer (NSCLC), accounting for approximately 40% of all lung cancer cases [1]. Despite significant advances in early detection and treatment strategies, LUAD remains a leading cause of cancer-related mortality globally [2,3]. Owing to its often asymptomatic nature in the early stages and the lack of effective biomarkers for prognosis, many patients are diagnosed at an advanced stage, which limits therapeutic options and impairs survival outcomes [4]. As such, there is a critical need for novel prognostic biomarkers to enhance early detection, predict patient outcomes, and guide therapeutic strategies.

The tumor microenvironment (TME) plays a pivotal role in the progression and metastasis of cancer, and cancer-associated fibroblasts (CAFs) are central components of this microenvironment [5]. CAFs, which are derived from normal fibroblasts or other stromal cells, are activated in response to tumorigenic signals and contribute to various aspects of cancer biology, including tumor growth, angiogenesis, immune evasion, and resistance to therapies [6,7]. CAFs play a pivotal role in the progression of lung cancer. Through their interactions with cancer cells, CAFs reprogram the metabolism of these cells and remodel the extracellular matrix (ECM), creating a supportive niche for cancer stem cells. CAFs induce collective invasion of tumor cells and shape the tumor immune microenvironment, facilitating both tumor metastasis and immune evasion [8]. CAFs secrete a variety of growth factors, cytokines, and extracellular matrix proteins that facilitate tumor cell migration, invasion, and metastasis [9]. Importantly, recent studies have demonstrated that CAFs not only influence tumor progression but are also linked to poor prognosis in several cancers, including lung adenocarcinoma [9,10,11].

The interaction between CAFs and tumor cells is mediated by a complex network of signaling molecules and gene expression changes. Many studies have identified specific CAF-related genes that contribute to tumor aggressiveness and metastasis. These genes include those involved in extracellular matrix remodeling, inflammatory responses, and immune cell recruitment, all of which play critical roles in tumor progression [12,13,14]. Importantly, recent research has suggested that CAF-related genes can serve as powerful prognostic biomarkers. In the context of LUAD, the expression levels of specific CAF-associated genes are correlated with poor survival, making them promising candidates for inclusion in prognostic models [15,16]. However, comprehensive models that integrate CAF-related gene signatures to predict clinical outcomes in patients with LUAD are lacking.

This study aimed to develop a prognostic model for lung adenocarcinoma (LUAD) patients on the basis of genes associated with tumor-associated fibroblast (CAF) infiltration scores while validating its accuracy and robustness. Key objectives include identifying CAF-related genes (CAFRGs), constructing a prognostic model, validating it in independent cohorts, assessing the clinical significance of risk scores, and conducting in vitro functional validation of key genes. These findings are intended to enhance precision in prognosis and therapeutic decision-making for patients with LUAD.

## 2. Results

### 2.1. Construction and Validation of the Prognostic Model Based on CAFRGs

This study utilized two LUAD cohorts from the TCGA and GEO databases, along with their corresponding clinical data. Table 1 and Appendix A summarize the demographic and clinical characteristics of the training, internal testing, and independent validation sets. After excluding samples with missing clinical information from the TCGA-LUAD dataset, a total of 504 patients with LUAD were included, of whom 183 were alive and 321 had died by the end of the follow-up period (median follow-up time: 1.789 years). This dataset was randomly divided into a training set (n = 353) and an internal testing set (n = 151) at a 7:3 ratio. As expected, no significant differences were observed in the major clinicopathological characteristics between the training, testing, and entire TCGA-LUAD cohorts (Table 1). Additionally, the study included the GEO dataset GSE31210, which comprises 226 patients with LUAD, with a mortality rate of 37.81% at the end of follow-up (median follow-up time: 4.720 years).

### 2.2. Construction and Validation of a Prognostic Model Based on CAF-Related Genes

On the basis of the TCGA LUAD dataset, CAF immune infiltration scores for each sample were obtained via the MCPCOUNTER and XCELL algorithms. Correlation analysis was performed to identify genes whose expression levels were correlated with CAF infiltration scores (CAFRGs, cancer-associated fibroblast-related genes; Appendix A). The intersection of these genes from the two algorithmic analyses revealed that 1154 genes were positively correlated with CAF infiltration and that 17 genes were negatively correlated (Appendix A). Using the training dataset, univariate Cox regression analysis identified a total of 174 CAFRGs that were associated with prognosis (Figure 1A). Gene selection was performed via least absolute shrinkage and selection operator (LASSO) regression, which identified 30 feature genes (Appendix A). These feature genes were further incorporated into a stepwise multivariate Cox regression analysis, and the results are shown in Figure 1B. The final prognostic risk model based on CAFRGs was constructed as follows: risk score = *COX6A1* Exp × (0.491) + *ENOX1* Exp × (0.409) + *FERMT2* Exp × (0.319) + *NID1* Exp × (0.257) + *LOX* Exp × (0.223) + *SNAI2* Exp × (0.148) + *GLI2* Exp × (0.137) + *ZNF154* Exp × (−0.136) + *COX7A1* Exp × (−0.179) + *NXPH3* Exp × (−0.182) + *FRMD4A* Exp × (−0.258) + *SYT11* Exp × (−0.391) + *ENTPD1* Exp × (−0.402). Receiver operating characteristic (ROC) curves demonstrated that the risk score had good predictive performance for patient prognosis, with area under the curve (AUC) values of 0.790, 0.819, and 0.839 for 1, 4, and 5 years, respectively (Figure 1C). High-risk patients had significantly poorer outcomes than low-risk patients (Figure 1D). In the training cohort, high-risk patients had shorter overall survival and more deaths than low-risk patients (Figure 1E,F). A heatmap was used to show the expression distribution of the genes in the model across high- and low-risk samples (Figure 1G).

### 2.3. Validation of the CAFRGs Risk Model

Next, the robustness of the model was validated using the entire TCGA LUAD dataset and an independent dataset, GSE31210. ROC curves were plotted on the basis of the risk scores for patient prognosis in the TCGA LUAD dataset, with a 1-year AUC value of 0.786 (Figure 2A). After the risk scores were divided into high- and low-risk groups on the basis of the median risk score, survival curve analysis indicated that patients in the low-risk group had a significantly better prognosis than those in the high-risk group (Figure 2B). The distribution of risk scores and survival outcomes revealed that low-risk patients had a lower mortality rate, whereas high-risk patients had a significantly higher mortality rate (Figure 2C). A heatmap revealed the expression patterns of CAFRGs across high- and low-risk samples in the TCGA LUAD dataset (Figure 2D). Similarly, in the GSE31210 dataset, ROC curves revealed the AUC values at 1–5 years, further validating the robustness of the model in an independent dataset (Figure 2E). Survival curve analysis revealed that high-risk patients had a worse prognosis (Figure 2F). Additionally, high-risk patients had shorter overall survival and more deaths than low-risk patients (Figure 2G). A heatmap demonstrated the differential expression of CAFRGs in high- and low-risk samples in the GSE31210 dataset (Figure 2H). In another independent cohort, GSE13213, the 1-year AUC of the ROC curve was 0.882 (Appendix A), and survival curve analysis similarly revealed poor prognosis in the high-risk group (Appendix A), with high-risk patients having shorter overall survival and more deaths (Appendix A). These results confirm the reliability and robustness of the CAFRG prognostic model.

### 2.4. CAFRG Risk Score as an Independent Prognostic Factor

In the TCGA LUAD dataset, univariate Cox regression analysis revealed that the CAFRG risk score was significantly associated with key clinical and pathological features, including distant metastasis, lymph node metastasis, invasion depth, and clinical stage, all of which were identified as risk factors (HR > 0, *p* < 0.05; Figure 3A). Further multivariate Cox regression analysis confirmed that the CAFRG risk score remained an independent prognostic factor (HR = 2.37, 95% CI: 2.16–4.04, *p* < 0.001; Figure 3B). In the GSE31210 dataset, univariate Cox analysis also revealed that the CAFRG risk score and clinical stage were significant factors affecting patient prognosis (HR > 0, *p* < 0.05; Figure 3C). A subsequent multivariate Cox regression further validated that the CAFRG risk score remained an independent prognostic factor (HR = 2.82, 95% CI: 1.23–6.04; *p* = 0.014; Figure 3D). These consistent results across multiple datasets suggest that the CAFRG risk score is a reliable and independent prognostic biomarker with substantial clinical value.

### 2.5. Construction and Evaluation of a Clinical Prediction Nomogram

A clinical prediction nomogram was constructed on the basis of the independent prognostic factors identified by multivariate Cox regression analysis in the TCGA LUAD dataset (Figure 4A). The accuracy of the nomogram was validated via calibration curves, receiver operating characteristic (ROC) curves, and decision curve analysis (DCA). The calibration curve demonstrated that the predicted 1-year, 3-year, and 5-year survival rates closely matched the actual survival rates, indicating excellent predictive performance (Figure 4B). ROC curve analysis revealed that the nomogram predicted the 1-, 3-, and 5-year survival probabilities, with AUC values of 0.811, 0.754, and 0.787, respectively (Figure 4C). The DCA results demonstrated that the nomogram provided a high net benefit across various risk thresholds, supporting its value in clinical decision-making (Figure 4D). For the GSE31210 dataset, a corresponding clinical prediction nomogram was constructed using the independent prognostic factors from the multivariate Cox regression analysis (Figure 4E). The calibration curve demonstrated a good fit between the predicted and actual survival outcomes (Figure 4F). ROC analysis revealed the ability of the nomogram to predict the 1-, 3-, and 5-year survival probabilities, further supporting its accuracy (Figure 4G). DCA indicated a high net benefit across risk thresholds, highlighting its potential for clinical application (Figure 4H). In summary, the nomograms constructed from both the TCGA LUAD and GSE31210 datasets displayed excellent predictive performance and clinical utility, making them valuable tools for personalized treatment and prognosis in patients with LUAD.

### 2.6. Construction of a Nomogram-Based Clinical Prediction Tool

Furthermore, we developed an online clinical prediction tool based on nomograms constructed from the TCGA LUAD and GSE31210 datasets (https://jingege.shinyapps.io/CAFRG_model/, accessed on 10 February 2025). This visualization tool allows users to make individualized survival predictions on the basis of various clinical features and risk scores. By adjusting clinical parameters, users can easily obtain survival curves and probabilities for individual patients. For example, when the parameters were set to T4, N0, and a risk score of 5, the predicted 1-year, 2-year, 3-year, and 5-year survival probabilities for patients were 84%, 66%, 49%, and 18%, respectively (Figure 5A). Figure 5B,C show the predicted survival curves for this setting, which indicate a progressively lower survival probability. On the other hand, when the parameters were set to T1, N0, and a risk score of 7, the predicted survival probabilities for 1-year, 2-year, 3-year, and 5-year survival were 47%, 17%, 5%, and 1%, respectively, suggesting poor long-term survival (Figure 5D,E).

### 2.7. Risk Score and Its Association with Immune Cell Infiltration and Immunotherapy

Using the TCGA LUAD dataset, we analyzed the correlation between the risk score and immune cell infiltration score calculated via the xCell algorithm. The results revealed a significant correlation between the risk score and the infiltration levels of several immune cell subsets, particularly classical dendritic cells (cDCs), M2 macrophages, hematopoietic stem cells (HSCs), and mast cells (Figure 6A). Additionally, the risk score was strongly correlated with the immune microenvironment score and immune score, with correlation coefficients of −0.445 and −0.435, respectively (Figure 6B,C). Further correlation analysis revealed significant associations between the risk score and the expression levels of multiple immune checkpoint genes (Figure 6D), especially *BTLA* (r = −0.330, Figure 6E) and *VSIR* (r = −0.311, Figure 6F). Moreover, via the tumor immune dysfunction and exclusion (TIDE) algorithm, we obtained immune infiltration scores for cancer-associated fibroblasts (CAFs) and myeloid-derived suppressor cells (MDSCs), along with T-cell dysfunction and exclusion scores. Subsequent correlation analysis revealed that the risk score was significantly correlated with T-cell dysfunction (Figure 6G) and exclusion (Figure 6H), as were the MDSC (Figure 6I) and CAF (Figure 6J) immune infiltration scores. Similarly, in the GSE31210 dataset, correlation analysis revealed associations with multiple immune cell infiltration scores (Appendix A) and T-cell dysfunction and exclusion scores on the basis of the TIDE algorithm (Appendix A). These results suggest that the risk score could serve as a potential indicator of the response of patients with LUAD to immunotherapy.

### 2.8. Risk Score and Its Association with Lung Adenocarcinoma Progression

We further assessed the correlation between the CAFRG risk score and the biological functions of LUAD. As shown in Figure 7A, the heatmap displays the correlation between the CAFRG risk score and oncogenes in the TCGA LUAD and GSE31210 datasets. These results indicate that the CAFRG risk score is significantly positively correlated with several known oncogenes, such as *PLK1*, *CDK1*, and *FOXM1*. Further correlation analysis revealed a significant association between the CAFRG risk score and sensitivity to various anticancer drugs, as calculated by the OncoPredict algorithm, including doramapimod, axitinib, urosertib, and niraparib (Figure 7B). Additionally, via gene set enrichment analysis (GSEA) of the TCGA LUAD dataset, we investigated the biological functions and signaling pathways associated with the CAFRG risk score. Biological function analysis revealed that the risk score was related to functions such as DNA replication, DNA double-strand break repair, cell adhesion regulation, and immune response activation (Figure 7C,D). In the signaling pathway analysis, the risk score was associated with several cancer-related signaling pathways, including the cell cycle, mismatch repair, DNA replication, the JAK-STAT signaling pathway, and cell adhesion molecules (Figure 7E,F).

### 2.9. COX6A1 as a Key Gene Promoting Tumor Progression in LUAD

In our analysis of the TCGA LUAD dataset, we identified *COX6A1* as a key gene that promotes tumor progression within our predictive model. Scatter plot analysis revealed a strong correlation between *COX6A1* expression and the risk score, suggesting that its expression is closely related to overall risk in patients with LUAD (Figure 8A). Further validation across multiple datasets revealed that *COX6A1* expression was significantly higher in tumor tissues than in normal tissues (Figure 8B). Survival analysis shows that LUAD patients with low expression of COX6A1 have a better prognosis (Figure 8C, Appendix A). To explore the role of *COX6A1* in the immune microenvironment, we evaluated the correlations between *COX6A1* expression and immune checkpoint gene expression, immune cell infiltration, and antitumor drug sensitivity. The results indicated that *COX6A1* expression was positively correlated with the expression of several immune checkpoint genes (Figure 8D). Moreover, *COX6A1* was also associated with immune cell infiltration scores for various cell types (Appendix A), including mast cells, tumor-associated fibroblasts (Figure 8E), and hematopoietic stem cells, as were matrix scores, immune microenvironment scores (Figure 8F), and immune scores. In terms of antitumor drug sensitivity, *COX6A1* expression was linked to increased sensitivity to drugs such as doramapimod, dihydrorotenone, and docetaxel (Figure 8G). Additionally, *COX6A1* expression was significantly positively correlated with tumor stemness, further suggesting its role in maintaining aggressive tumor characteristics (Figure 8H). Finally, gene set enrichment analysis (GSEA) revealed that *COX6A1* expression was associated with multiple important signaling pathways, such as the TGF-β and JAK-STAT pathways, and with biological functions related to DNA replication, oxidative phosphorylation, cell adhesion molecules, and base excision repair (Figure 8I–K). These findings highlight *COX6A1* as a potential therapeutic target and a critical regulator of the tumor immune microenvironment and drug sensitivity in LUAD.

### 2.10. COX6A1 Is a Gene That Promotes Tumor Progression in the Model

CCK8 assays revealed that silencing *COX6A1* significantly inhibited the proliferation of the lung adenocarcinoma cell lines A549 (Figure 9A) and H1299 (Figure 9B). Figure 9C,D presents the dose–response curves for A549 and H1299 cells treated with various concentrations of doramapimod, demonstrating that *COX6A1* knockdown increased their sensitivity to the drug, as reflected by a reduction in the IC50 values. Transwell migration assays revealed that *COX6A1* knockdown significantly inhibited the migration capacity of A549 and H1299 cells (Figure 9E,F). Similarly, EdU assays demonstrated that *COX6A1* knockdown markedly reduced cell proliferation (Figure 9G,H). Further analysis via β-galactosidase staining revealed that silencing *COX6A1* induced senescence in lung cancer cells (Figure 9I,J). Quantitative PCR (qPCR) analysis confirmed that *COX6A1* expression was significantly reduced after *COX6A1* knockdown (Figure 9K). Moreover, the epithelial marker CDH1 and the senescence-associated genes *CDKN1A* (P21) and *CDKN2A* (P16) were significantly upregulated, whereas the mesenchymal marker CDH2 was significantly downregulated (Figure 9L). At the protein level, Western blot analysis revealed consistent changes in protein expression, corroborating the mRNA data and further validating the critical role of *COX6A1* in regulating tumor cell migration and senescence (Figure 9M,N).

### 2.11. COX6A1 Knockdown in Lung Cancer Cells Promotes CAF Infiltration

Further in vitro studies were conducted to investigate the effect of *COX6A1* overexpression in lung cancer cells on the infiltration of CAFs. Analysis of the TCGA LUAD dataset (Figure 10A) and multiple LUAD datasets from the GEO database (Figure 10B) revealed a significant correlation between *COX6A1* expression and the expression of CAF activation-related cytokines. To validate this finding, we used qPCR to measure the expression levels of several CAF-related cytokines in lung cancer cells with *COX6A1* knockdown. The results showed a significant decrease in the expression of *TGFB2* (Figure 10C), *CXCL12* (Figure 10D), and *FGF2* (Figure 10E). Additionally, ELISA further confirmed that the expression level of CXCL12 in the culture supernatant of *COX6A1*-knockdown lung cancer cells was significantly reduced (Figure 10F). To investigate the specific mechanism by which *COX6A1* affects CAF infiltration, we established a co-culture system of lung cancer cells and human embryonic lung cells WI-38 (Figure 10G). qPCR analysis of CAF marker gene expression in co-cultured WI-38 cells showed a significant upregulation of *α-SMA* (Figure 10H), *FN1* (Figure 10I), and *VIM* (Figure 10J) RNA levels. Immunofluorescence analysis further demonstrated a significant increase in α-SMA expression in WI-38 cells after co-culture with lung cancer cells (Figure 10K), with consistent results obtained through quantitative analysis (Figure 10L). Moreover, Transwell migration assays indicated that the migratory capacity of WI-38 cells was significantly enhanced in the co-culture system (Figure 10M), as confirmed by quantitative analysis (Figure 10N). These results suggest that *COX6A1* overexpression promotes the expression of CAF-related cytokines, enhancing the attraction of lung cancer cells to CAFs, thereby facilitating CAF infiltration.

## 3. Discussion

This study systematically analyzed tumor-associated fibroblast (CAF) immune infiltration-related genes and successfully constructed and validated a prognostic model for lung adenocarcinoma (LUAD), revealing the critical role of CAFs in the tumor immune microenvironment and providing new insights and theoretical foundations for precision medicine. We found that this model not only effectively predicts the prognosis of patients with LUAD but is also closely associated with the tumor immune microenvironment, drug sensitivity, and tumor biological characteristics, such as proliferation, migration, and senescence. These findings provide important clues for a deeper understanding of the mechanisms underlying CAFs in LUAD and for the development of new therapeutic strategies.

One of the key findings of this study is that the CAF-related gene (CAFRG) risk score is significantly associated with immune microenvironment scores and immune cell infiltration, particularly with the expression of immune checkpoint genes. Changes in the immune microenvironment play crucial roles in tumor immune escape [17,18], and CAFs, as important components of the tumor microenvironment, may influence tumor immune escape mechanisms by modulating immune cell infiltration and immune checkpoint activation [19,20]. We observed that the CAFRG risk score was correlated with the infiltration of various immune cells and was inversely related to the immune microenvironment score. These results suggest that high-risk patients have a lower immune microenvironment and immune activity, which may indicate immune escape features. Using the TIDE tool (http://tide.dfci.harvard.edu/, accessed on 10 January 2025), we found that the risk score was negatively correlated with T-cell dysfunction and positively correlated with T-cell exclusion, suggesting that low-risk patients may have a more active T-cell immune response and an immune system capable of effectively fighting the tumor, whereas high-risk patients may face immune escape challenges, as reflected by impaired T-cell function and T-cell exclusion, leading to weakened immune surveillance [21]. Immune checkpoint genes (e.g., *PDCD1* and *CTLA4*) are often closely associated with immune escape mechanisms, as tumor cells express immune checkpoint molecules to suppress immune system attacks, thereby evading host immune surveillance [22,23,24]. We found that the risk score was negatively correlated with the expression of these immune checkpoint genes, further suggesting that the CAFRG risk score could predict immune escape and the immune therapy response in patients with LUAD. Moreover, we explored the relationship between the risk score and tumor biological functions and found that it is closely related to signaling pathways and biological functions involved in LUAD proliferation, migration, and immune system activation. Drug resistance has become a major challenge in clinical cancer therapy. Resistance not only significantly reduces treatment efficacy but also worsens patient prognosis, increasing the complexity and cost of treatment [25]. The risk score was also found to be closely associated with antitumor drug sensitivity, particularly with a significant negative correlation with the sensitivity to doramapimod (a MAPK/ERK inhibitor). In summary, our study elucidates the comprehensive role of CAF-related genes in modulating the immune microenvironment, influencing immune escape mechanisms, and impacting cellular pathways related to tumor growth and drug resistance. Future studies should focus on validating these findings in larger, independent cohorts and exploring the therapeutic potential of targeting CAFs and associated pathways in lung adenocarcinoma treatment.

Furthermore, we identified a key gene in the model, cytochrome c oxidase subunit 6A1 (*COX6A1*). COX6A1 is a mitochondrial membrane protein that is widely involved in cellular energy metabolism and plays an important role in oxidative phosphorylation (OXPHOS) in particular [26]. OXPHOS is the main pathway for ATP production, and tumor cells require large amounts of energy during rapid proliferation and in response to external stress; thus, OXPHOS plays a crucial role in tumor progression [27,28]. Although the Warburg effect dominates energy metabolism in some tumors, recent studies have shown that OXPHOS is significantly activated in chemoresistant and cancer stem cells, contributing to tumor cell survival and metastasis [29]. Therefore, COX6A1 may serve as a new target for anticancer therapies. Our study further revealed that the upregulation of *COX6A1* is associated with the degree of immune cell infiltration in tumor tissues, particularly with the expression of immune checkpoint genes, suggesting that *COX6A1* may affect tumor immune escape by modulating immune cell function or through immune checkpoint pathways. In vitro experiments confirmed *COX6A1’s* role in lung cancer cells. *COX6A1* knockdown inhibited tumor cell migration and proliferation while promoting cellular senescence—a state where cells cease to divide after stress, accompanied by metabolic and gene expression changes. Tumor cell senescence can suppress tumor development but also influence metastasis potential and immune escape [30,31,32]. Knockdown of *COX6A1* significantly upregulated senescence-associated genes, such as *CDKN1A* and *CDKN2A*, indicating that *COX6A1* may inhibit tumor progression by inducing senescence. Additionally, it is important to discuss the role of extracellular matrix remodeling and CAFs in the regulation of cancer aggressiveness, as these factors play crucial roles in the tumor microenvironment. ECM remodeling and interactions with CAFs can significantly influence tumor behavior, including proliferation, migration, and immune responses [33,34,35].

We conducted further in vitro studies to investigate the impact of *COX6A1* knockdown on CAF infiltration in lung cancer cells. Analysis of multiple lung adenocarcinoma datasets revealed a significant correlation between *COX6A1* expression and the expression of CAF activation-related cytokines. Our in vitro experiments demonstrated that *COX6A1* knockdown in lung cancer cells led to the upregulation of *TGFB2*, *CXCL12*, and *FGF2*. Further, co-culture experiments with lung cancer cells and human embryonic lung WI-38 cells showed increased expression of *α-SMA*, *FN1*, and *VIM* in WI-38 cells, along with significantly enhanced migration capacity. These results indicate that *COX6A1* knockdown promotes the expression of CAF-related cytokines, thereby enhancing the attraction of CAFs to lung cancer cells and promoting CAF infiltration. Interestingly, CAFs typically support tumor growth and metastasis in the tumor microenvironment by secreting cytokines and matrix remodeling factors [5,9]. This appears to be contrary to the high expression of *COX6A1* in lung cancer, prompting us to further investigate this issue through rigorous experiments.

Overall, this study constructed a prognostic model based on CAF immune infiltration-related genes, providing an effective tool for prognostic evaluation in patients with LUAD and opening new directions for personalized treatment of LUAD. In particular, we found that the upregulation of *COX6A1* in LUAD is closely associated with immune cell infiltration, immune escape, and biological features such as tumor cell migration and senescence, suggesting its important regulatory role in the tumor microenvironment. Our study deepens the understanding of the role of CAFs in tumor progression and highlights the importance of *COX6A1* in the tumor immune microenvironment, revealing its clinical application prospects as a potential therapeutic target.

## 4. Materials and Methods

### 4.1. Dataset

This study utilized lung adenocarcinoma (LUAD) data from multiple public databases. Initially, the gene expression data and clinical information of patients with LUAD were obtained from the Cancer Genome Atlas (TCGA) database. The TCGA dataset included gene expression data from 572 patients, comprising 513 tumor tissues and 59 adjacent normal tissues, along with corresponding clinical characteristics (e.g., age, sex, and tumor stage). To validate the model’s efficacy, the GSE31210 dataset was also extracted from the Gene Expression Omnibus (GEO) database, which contains gene expression and prognosis information for 226 primary LUAD samples.

### 4.2. Immune Infiltration Analysis

To evaluate the immune microenvironment in LUAD samples, particularly the infiltration level of CAFs, immune infiltration analysis was conducted via the xCell [36] and TIMER [37] algorithms. xCell Algorithm: xCell is a high-resolution immune cell infiltration analysis tool that quantitatively predicts immune cell infiltration in the TME. Through xCell analysis, infiltration scores for CAFs and other immune and stromal cells in tumor samples were obtained. TIMER Algorithm: TIMER is an online platform for immune infiltration analysis that quantifies the infiltration levels of various immune cells (including T cells, B cells, and macrophages) and provides T-cell dysfunction and exclusion scores for each sample.

### 4.3. Prognostic Model Construction and Validation

All genes associated with CAF scores (|correlation coefficient| > 0.25, *p* < 0.05) in the TCGA LUAD dataset were analyzed via the Spearman method and defined as CAF-related genes (CAFRGs). The CAFRG prognostic model was constructed and validated via dataset splitting. The TCGA LUAD dataset was split into a training set and an internal test set at a 7:3 ratio. Clinical characteristic analysis: Chi-square tests were used to analyze the differences in the distributions of clinical characteristics (e.g., sex, age, tumor stage, and lymph node metastasis) between the training and validation sets. Univariate Cox regression analysis: Each CAFRG was evaluated for its relationship with overall survival (OS), and the hazard ratio (HR) and corresponding *p* value were calculated. CAFRGs with *p* values less than 0.05 were selected as candidate genes. LASSO regression analysis: To reduce model complexity and select the best prognostic factors, the least absolute shrinkage, and selection operator (LASSO) regression analysis was conducted to screen the most OS-related CAFRGs from the univariate Cox regression. Multivariate Cox regression analysis: Based on the LASSO regression results, multivariate Cox stepwise regression analysis was further performed to establish a prognostic model for CAF-related genes. Each patient’s risk score was calculated from the Cox regression results via the formula $\mathrm{Risk}\;\mathrm{score}\;=\sum_{i=1}^{n}{Exp}_{i}\times {coef}_{i}$, where ${Exp}_{i}$ is the expression value of the *i*th ARG and ${coef}_{i}$ is the Cox regression coefficient of that gene. Model validation: The predictive performance of the prognostic model was evaluated via Kaplan–Meier survival analysis and time-dependent receiver operating characteristic (ROC) curves (using the “TimeROC” package, version 0.4) in the training set, internal test set, and validation set.

### 4.4. Nomogram Construction and Evaluation

On the basis of the results of multivariate Cox regression analysis, a nomogram was constructed to provide an individualized survival prediction tool for patients. The nomogram combined the CAFRG risk score with independent prognostic factors identified from the multivariate Cox analysis to generate a survival probability prediction for patients. Nomogram Construction: Using the “rms” package (version 6.9-0), a comprehensive model was constructed that visually represents the impact of risk scores and clinical characteristics on patient survival. Model evaluation: The nomogram’s predictive performance was assessed by calculating the C-index (concordance index). Calibration curves were used to evaluate the model’s applicability in clinical practice. Decision curve analysis (DCA) was employed to assess the clinical decision value of the nomogram.

### 4.5. Clinical Prediction Tool Development

On the basis of the prognostic model and nomogram, a clinical prediction tool was developed to assist clinicians in predicting patient survival risk on the basis of clinical characteristics and CAFRG risk scores. This tool can generate survival curves and predict 1-year, 3-year, and 5-year survival probabilities on the basis of patient clinical information and risk scores. When implemented via the R Shiny package (version 1.10.0), the tool allows clinicians to easily utilize it, aiding in the formulation of more personalized treatment plans. This tool can improve the accuracy of prognosis assessment for patients with cancer and guide clinical decision-making.

### 4.6. Drug Sensitivity Analysis

The purpose of drug sensitivity analysis is to evaluate the sensitivity of various tumor samples to multiple antitumor drugs, providing a basis for clinical treatment. Drug sensitivity data were obtained from the GDSC database (https://www.cancerrxgene.org/, accessed on 28 December 2024), which offers cell line sensitivity data for a range of drugs, including chemotherapeutic and targeted agents [38]. Drug sensitivity prediction for each sample was conducted via the OncoPredict package (version 1.2) [39].

### 4.7. GSEA Enrichment Analysis

Gene set enrichment analysis (GSEA) was employed to investigate signaling pathways and biological functions associated with risk scores. Initially, all genes correlated with the risk scores were identified through correlation analysis and ranked according to their correlation coefficients. GSEA was conducted via the “ClusterProfiler” R package (version 4.14.3), which is based on predefined C2 (curated gene sets) and C5 (GO gene sets) [40]. Pathways significantly associated with risk scores were identified on the basis of the normalized enrichment score (NES) and false discovery rate (FDR).

### 4.8. Cell Culture

The cell lines used in this study included the human non-small cell lung cancer (NSCLC) cell lines A549 and H1299. All the cell lines were purchased from the American Type Culture Collection (ATCC) and cultured under standard conditions. A549 and H1299 cells were cultured in RPMI-1640 medium supplemented with 10% fetal bovine serum (FBS, Biological Industries, Beit - Haemek, Israel). The cells were maintained at 37 °C in a humidified incubator (Thermo Fisher Scientific, Waltham, MA, USA) with 5% CO_2_ and passaged when they reached 70–80% confluence. Morphological examination confirmed the absence of contamination in all the cell lines before use.

### 4.9. shRNA Construction and Transfection

To investigate the function of the key gene *COX6A1* in the model, we designed shRNA expression vectors targeting *COX6A1*. The specific steps are as follows:

shRNA Design: Specific shRNA sequences targeting *COX6A1* were designed via siRNA design software (e.g., BLOCK-iT™ RNAi Designer, https://rnaidesigner.thermofisher.com/rnaiexpress/, accessed on 18 December 2024).

Vector construction: The shRNA sequences were subsequently cloned and inserted into the pGreen vector, and successful cloning was confirmed via agarose gel electrophoresis.

Cell Transfection: A549 or H1299 cells at 70–80% confluence were cotransfected with the shRNA vector via the Lipofectamine 3000 transfection reagent (Thermo Fisher Scientific, Waltham, MA, USA). Subsequent experimental analyses were conducted 48 h posttransfection.

Transfection efficiency: Transfection efficiency was assessed via qPCR and Western blot analysis.

### 4.10. CCK-8 Assay

Cell proliferation was assessed via the Cell Counting Kit-8 (CCK-8) assay (Beyotime, Shanghai, China). The experimental steps were as follows: transfected A549 or H1299 cells were seeded into 96-well plates at a density of 2 × 10^3^ cells per well. After 24 h of culture, various concentrations of drugs or media were added. At 24, 48, and 72 h posttreatment, 10 μL of CCK-8 solution was added to each well, and the cells were incubated for an additional 2 h. The optical density (OD) at 450 nm was measured via a microplate reader (BioTek, Winooski, VT, USA). The cell proliferation inhibition rate was calculated on the basis of the OD values to generate proliferation curves for different time points and drug concentrations.

### 4.11. Doramapimod Dose–Response Curve

To evaluate the inhibitory effect of Doramapimod on cells, a dose–response curve was constructed.

Transfected A549 or H1299 cells were seeded into 96-well plates, with drug concentrations ranging from 0 to 512 μM. The effects of various drug concentrations on cell proliferation were assessed 24 h posttreatment via the CCK-8 assay. Dose–Response Analysis: Dose–response curves were generated via GraphPad V8.3.0 software.

### 4.12. Transwell Cell Migration Assay

Cell migration ability was assessed via a Transwell chamber assay. The experimental procedure was as follows: transfected A549 or H1299 cells were seeded into the upper chamber with serum-free medium, and the lower chamber contained medium supplemented with 20% FBS as a chemoattractant. After 24 h of incubation, nonmigrated cells in the upper chamber were removed with a sterile cotton swab. The migrated cells in the lower chamber were fixed and stained with crystal violet. The stained cells were observed and counted under a microscope (Olympus, Tokyo, Japan). The results are expressed as the number of migrated cells, and differences between groups were compared.

### 4.13. EdU Cell Proliferation Assay

Cell proliferation was evaluated via an EdU (5-ethynyl-2’-deoxyuridine) assay kit (Beyotime, Shanghai, China). The experimental steps were as follows: transfected A549 or H1299 cells were seeded into 96-well plates, treated with drugs, and then incubated with EdU labeling solution for 2 h. After fixation, the cells were stained via an EdU staining kit. EdU-positive cells were counted under a fluorescence microscope (Nikon, Tokyo, Japan), and the proliferation index was calculated.

### 4.14. β-Galactosidase Assay

To assess cellular senescence, a β-galactosidase staining kit (Novozam, Nanjing, China) was used. The experimental procedure was as follows: Transfected cells were seeded in culture dishes and treated with specific drugs or stimuli. β-Galactosidase staining was performed according to the kit instructions. Senescent cells, which appeared blue under a microscope, were observed and quantified. The proportion of senescent cells was calculated and compared between groups.

### 4.15. Quantitative PCR (qPCR)

Quantitative PCR (qPCR) was used to measure the relative expression levels of the *CDKN1A*, *CDKN2A*, *CDH1*, and *CDH2* genes, with *GAPDH* serving as the internal control. Total RNA was extracted and reverse-transcribed into cDNA via reverse transcriptase. Each qPCR mixture included SYBR Green Master Mix, gene-specific primers (*CDKN1A*_F: TGTCCGTCAGAACCCATGC, *CDKN1A*_R: AAAGTCGAAGTTCCATCGCTC, *CDKN2A*_F: GGGTTTTCGTGGTTCACATCC, CDKN2A_R: CTAGACGCTGGCTCCTCAGTA, *CDH1*_F: CGAGAGCTACACGTTCACGG, *CDH1*_R: GGGTGTCGAGGGAAAAATAGG, *CDH2*_F: AGCCAACCTTAACTGAGGAGT, *CDH2*_R: GGCAAGTTGATTGGAGGGATG, *GAPDH*_F: ACAACTTTGGTATCGTGGAAGG, *GAPDH*_R: GCCATCACGCCACAGTTTC), and a cDNA template. The PCR program was set as follows: initial denaturation at 95 °C for 30 s, followed by 40 cycles of 95 °C for 5 s and 60 °C for 30 s. Relative expression levels were calculated via the 2^−ΔΔCt^ method, where ΔCt represents the difference in Ct values between the target gene and *GAPDH* and where ΔΔCt represents the difference between the experimental and control groups.

### 4.16. Western Blotting

Western blotting was used to detect the protein expression of CDKN1A, CDKN2A, CDH1, and CDH2, with GAPDH serving as the internal control. The protein concentration in the cell lysates was determined via the BCA method, and 30 μg of protein was loaded for SDS-PAGE. After transfer, the membranes were blocked with 5% nonfat milk for 1 h and incubated with specific primary antibodies overnight. All primary antibodies (CDKN1A, CDKN2A, CDH1, CDH2, and GAPDH) were purchased from Proteintech and used at a dilution of 1:2000. The catalog numbers for the antibodies were as follows: CDKN1A (10355-1-AP), CDKN2A (10883-1-AP), CDH1 (20874-1-AP), CDH2 (22018-1-AP), and GAPDH (60004-1-Ig). The membranes were subsequently incubated with HRP-conjugated secondary antibodies and developed via enhanced chemiluminescence (ECL) reagents. The signals were captured via a chemiluminescence imaging system, and protein expression levels were quantified via ImageJ software (version 2.16.0), with GAPDH used as the normalization control.

### 4.17. ELISA (Enzyme-Linked Immunosorbent Assay)

In this study, we used ELISA to quantify the levels of CXCL12 in the culture supernatant of *COX6A1* knockdown lung cancer cells. We collected the culture supernatant from *COX6A1* knockdown-treated lung cancer cells and processed it according to the manufacturer’s instructions. ELISA plates were coated with CXCL12-specific antibodies. After washing, samples were added and incubated at appropriate temperatures and times to ensure binding. Subsequently, an enzyme-labeled secondary antibody (Beyotime, Shanghai, China) was added, followed by another wash and the addition of a substrate solution to generate a measurable signal. The concentration of CXCL12 in the samples was calculated by comparing the amount of product from the enzymatic reaction to a standard curve.

### 4.18. Co-Culture System

To explore the impact of COX6A1 on lung cancer cell-induced CAF infiltration, we established a co-culture system with lung cancer cells and human embryonic lung WI-38 cells. Lung cancer cells were first seeded in appropriate media to reach 70–80% confluence. WI-38 cells were then introduced at a suitable ratio to ensure effective contact and interaction between the cells. The media were regularly changed to remove metabolic waste and maintain optimal growth conditions.

### 4.19. Immunofluorescence Analysis

We used immunofluorescence to detect α-SMA expression in WI-38 cells. Co-cultured WI-38 cells were fixed with 4% paraformaldehyde and permeabilized to allow antibody access. Cells were incubated with an α-SMA-specific primary antibody, followed by PBS washing to remove unbound antibodies. A fluorescently labeled secondary antibody was then added, incubated, and washed to remove excess secondary antibody. Cells were observed using a fluorescence microscope, and images were captured for quantitative analysis of α-SMA expression.

### 4.20. Statistical Analysis

Statistical analyses were conducted via GraphPad V8.3.0 software (GraphPad Software, LLC, San Diego, CA, USA). The data are presented as the means ± standard deviations. Differences between the two groups were assessed via Student’s *t*-test. All the statistical tests were two-tailed, with a *p* value of <0.05 considered statistically significant.

## Figures and Tables

**Figure 1 ijms-26-03478-f001:**
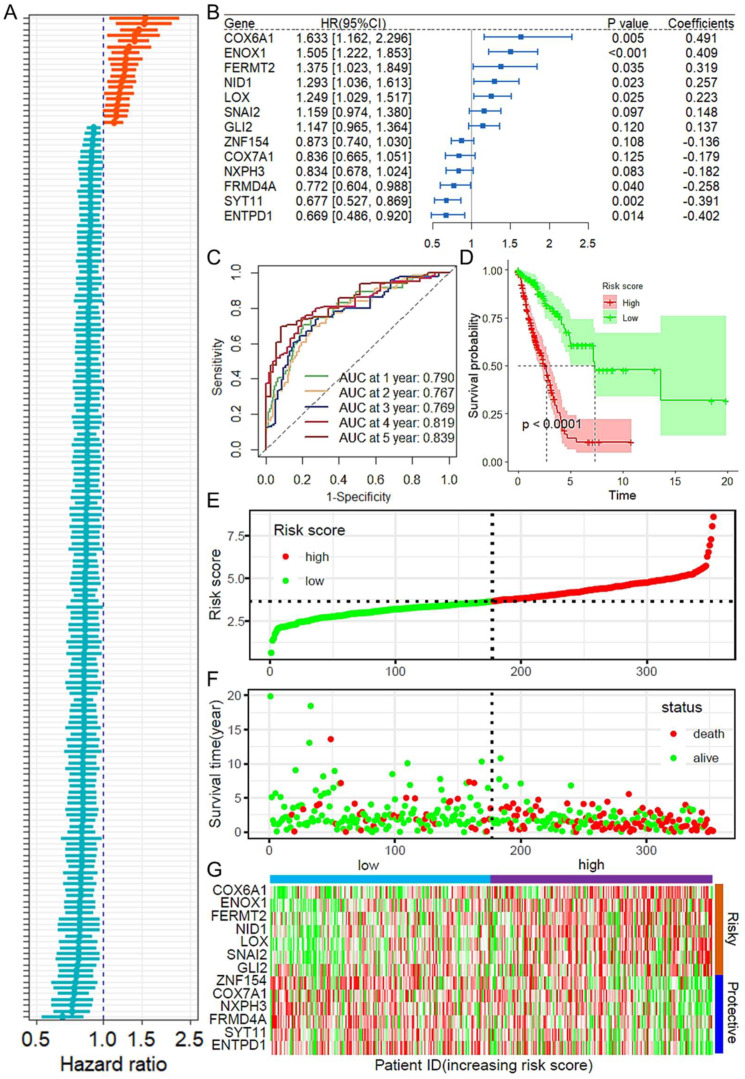
Construction and performance analysis of the CAFRG prognostic model. (**A**) Univariate Cox regression analysis identified CAFRGs associated with prognosis in the training cohort. (**B**) Multivariate Cox regression analysis was used to construct the prognostic model. (**C**) Time-dependent ROC curve showing the ROC and AUC values for 1–5 years in the training cohort. (**D**) Survival curves of high-risk and low-risk patients in the training cohort. (**E**) Distribution of risk scores in high- and low-risk patients and their survival outcomes and times in the training cohort. (**F**) Distribution of survival outcomes and times for high- and low-risk patients. (**G**) Heatmap showing the differential expression of CAFRGs in high- and low-risk samples in the training cohort. The gradient from green to red represents the range of expression in the dataset. ROC: receiver operating characteristic; CAFRG: cancer-associated fibroblast-related genes; AUC: area under the curve.

**Figure 2 ijms-26-03478-f002:**
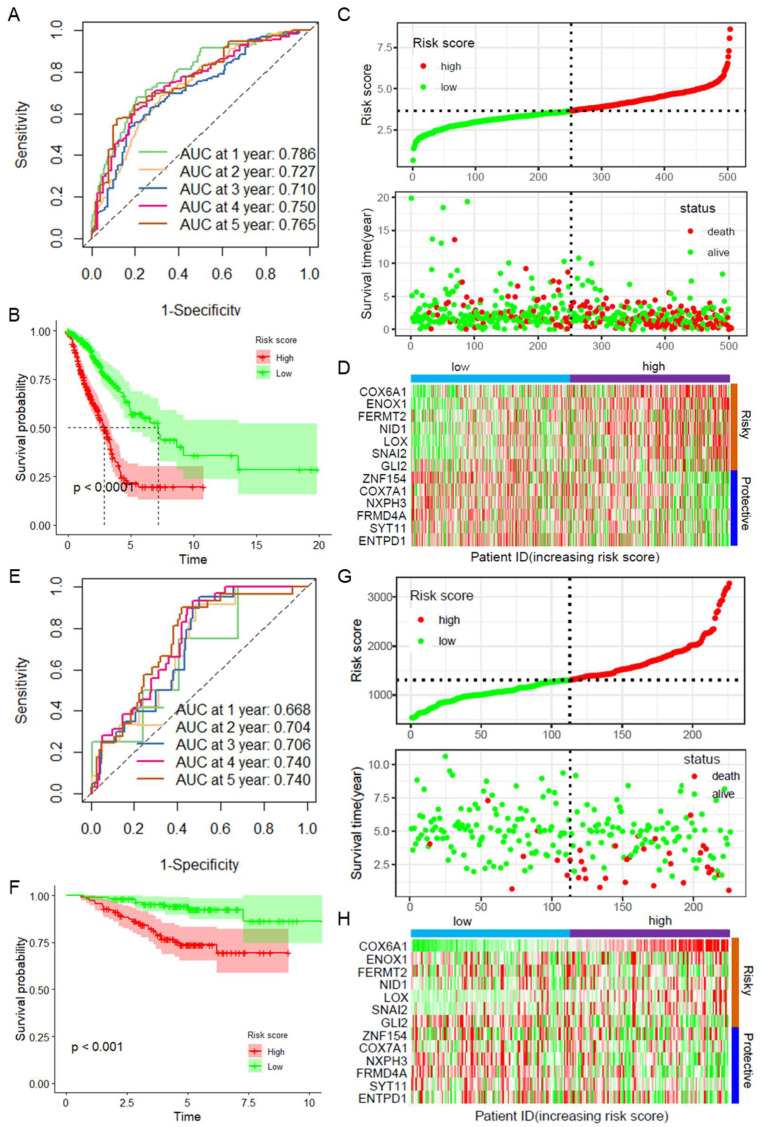
Validation of the model’s robustness in the TCGA LUAD and GSE31210 datasets. (**A**) Time-dependent ROC curve showing ROC and AUC values for 1–5 years in the TCGA LUAD dataset; (**B**) survival curves comparing high-risk and low-risk patients in the TCGA LUAD dataset; (**C**) distribution of risk scores and survival outcomes in high- and low-risk samples in the TCGA LUAD dataset; (**D**) heatmap showing the differential expression of CAFRGs in high- and low-risk samples in the TCGA LUAD dataset. The gradient from green to red represents the range of expression in the dataset; (**E**) time-dependent ROC curve showing ROC and AUC values for 1–5 years in the GSE31210 dataset; (**F**) survival curve comparison between high-risk and low-risk patients in the GSE31210 dataset; (**G**) distribution of risk scores and survival outcomes in high- and low-risk patients in the GSE31210 dataset; (**H**) heatmap showing the expression patterns of CAFRGs in high- and low-risk samples in the GSE31210 dataset. The gradient from green to red represents the range of expression in the dataset. TCGA: The Cancer Genome Atlas; LUAD: Lung adenocarcinoma; ROC: Receiver operating characteristic; CAFRG: Cancer-associated fibroblast-related genes; AUC: Area under the curve.

**Figure 3 ijms-26-03478-f003:**
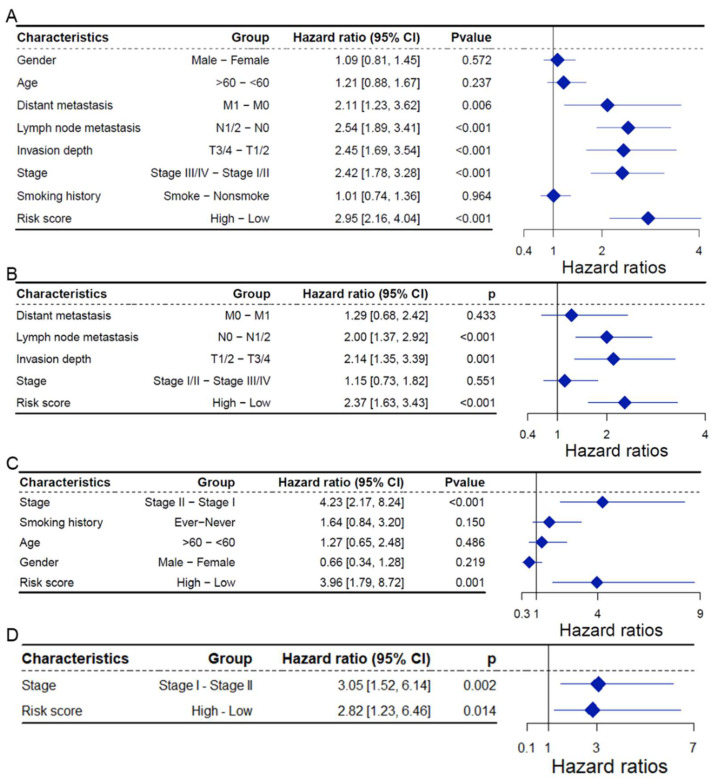
CAFRG risk score as an independent prognostic factor. (**A**) Forest plot showing univariate Cox analysis results of the CAFRG risk score and major clinicopathological features for patient prognosis in the TCGA LUAD dataset. (**B**) Forest plot showing the multivariate Cox analysis results of significant factors from univariate analysis for patient prognosis in the TCGA LUAD dataset. (**C**) Forest plot showing univariate Cox analysis results of the CAFRG risk score and major clinicopathological features for patient prognosis in the GSE31210 dataset. (**D**) Forest plot showing the multivariate Cox analysis results of significant factors from univariate analysis for patient prognosis in the GSE31210 dataset. TCGA: The Cancer Genome Atlas; LUAD: Lung adenocarcinoma; CAFRG: Cancer-associated fibroblast-related genes.

**Figure 4 ijms-26-03478-f004:**
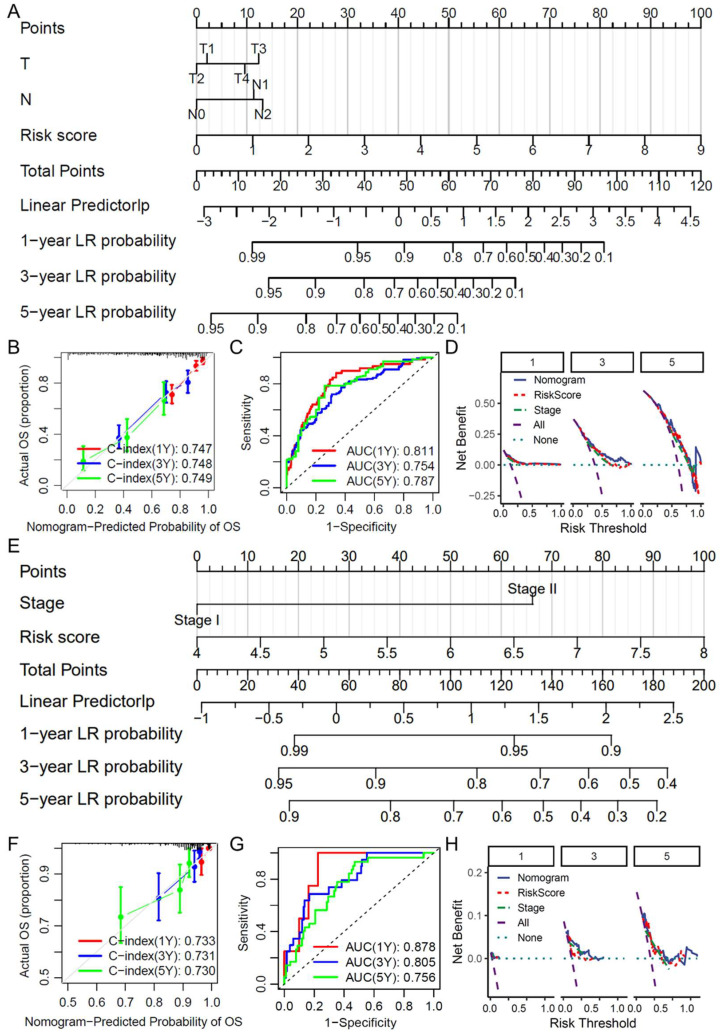
Construction and evaluation of the clinical prediction nomogram. (**A**) Clinical prediction nomogram constructed on the basis of independent prognostic factors identified by multivariate Cox analysis in the TCGA LUAD dataset; (**B**) Calibration curve showing the agreement between the predicted and actual 1-, 3-, and 5-year survival rates for the TCGA LUAD nomogram; (**C**) ROC curve evaluating the accuracy of the TCGA LUAD nomogram for predicting 1-, 3-, and 5-year survival; (**D**) DCA curve for the TCGA LUAD nomogram showing net benefit across different risk thresholds; (**E**) Clinical prediction nomogram constructed for the GSE31210 dataset on the basis of independent prognostic factors from multivariate Cox analysis; (**F**) Calibration curve for the GSE31210 nomogram; (**G**) Kaplan–Meier survival curve for the GSE31210 nomogram; (**H**) Decision curve analysis for the GSE31210 nomogram. TCGA: The Cancer Genome Atlas; LUAD: Lung adenocarcinoma; ROC: Receiver operating characteristic; AUC: Area under the curve; DCA: Decision curve analysis; Cox: Cox regression analysis.

**Figure 5 ijms-26-03478-f005:**
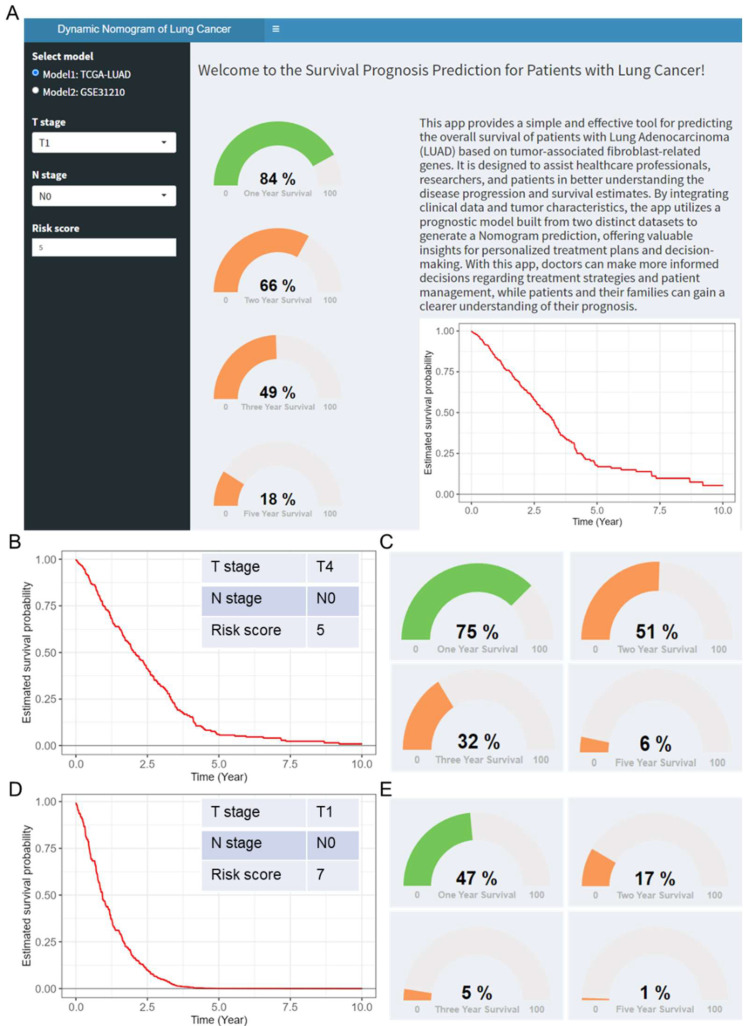
Construction of the nomogram-based clinical prediction tool. (**A**) An online tool designed for visual prediction based on the TCGA LUAD and GSE31210 data predicted survival probabilities and curves for parameters T4, N0, and risk score of 5 (**B**,**C**); Predicted survival probabilities and curves for parameters T1, N0, and risk score of 7 (**D**,**E**).

**Figure 6 ijms-26-03478-f006:**
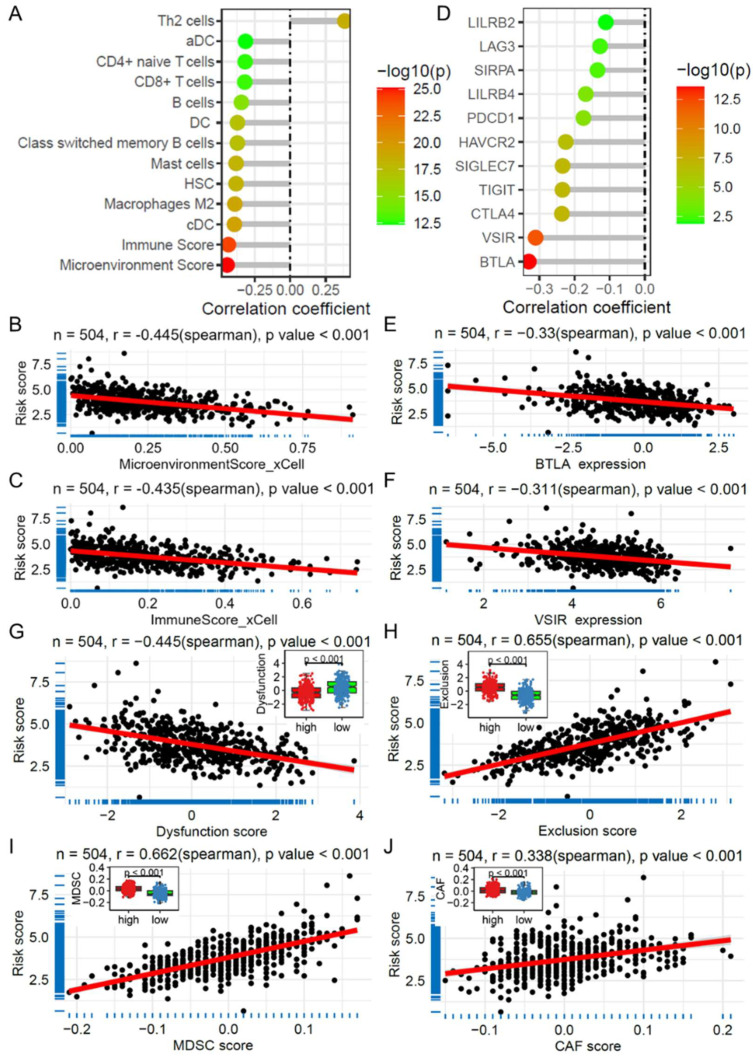
Correlation of the risk score with immune cell infiltration and immunotherapy in the TCGA LUAD dataset. (**A**) Lollipop plot showing the correlation between the risk score and immune cell infiltration score calculated via the xCell algorithm. Scatter plots depicting the correlation between the risk score and immune microenvironment score (**B**) and immune score (**C**). (**D**) Lollipop plot illustrating the correlation between the risk score and expression of immune checkpoint genes. Scatter plots showing the correlation between the risk score and the expression of *BTLA* (**E**) and *VSIR* (**F**). Scatter plots and box plots demonstrate the correlation between the risk score and immune infiltration score of T-cell dysfunction (**G**), exclusion (**H**), MDSC (**I**), and CAF (**J**) based on the TIDE algorithm. TCGA: The Cancer Genome Atlas; LUAD: Lung adenocarcinoma; TIDE: Tumor immune dysfunction and exclusion; CAF: Cancer-associated fibroblasts; MDSC: Myeloid-derived suppressor cells.

**Figure 7 ijms-26-03478-f007:**
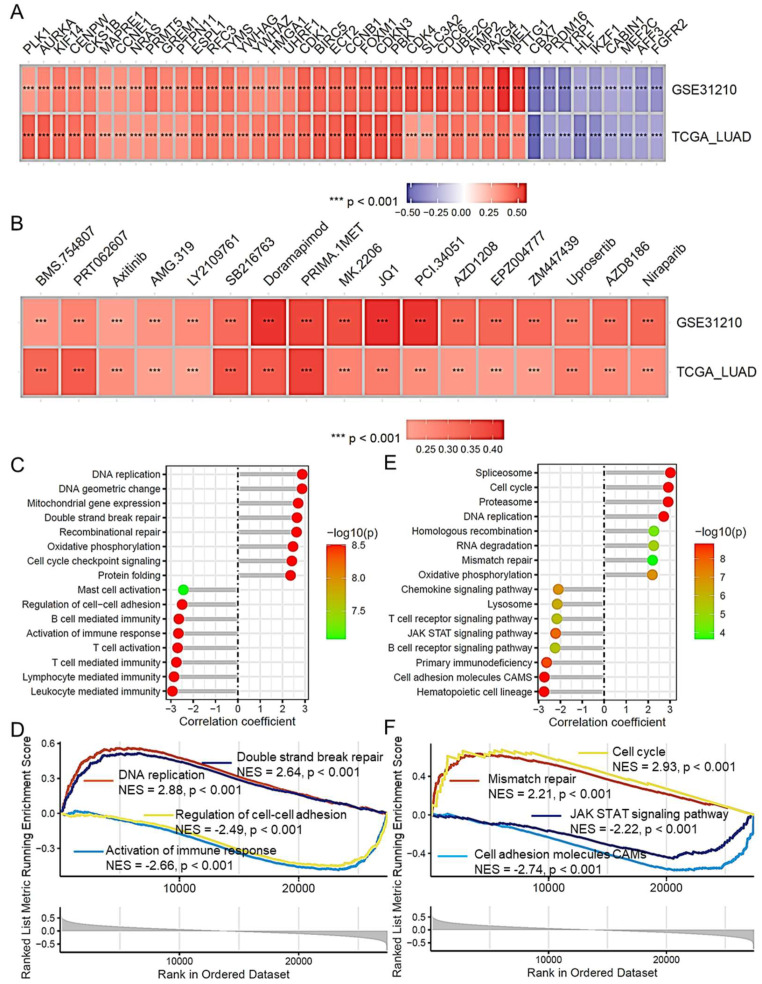
Correlation between the CAFRG risk score and progression of lung adenocarcinoma. Heatmap displaying the correlation between the CAFRG risk score and oncogenes in the TCGA LUAD and GSE31210 datasets (**A**) and the association with anticancer drug sensitivity calculated via OncoPredict (**B**). Lollipop plots showing the results of GSEA enrichment analysis for biological functions (**C**) and signaling pathways (**E**) in the TCGA LUAD dataset. GSEA plots show enrichment of the CAFRG risk score in key biological functions (**D**) and signaling pathways (**F**). TCGA: The Cancer Genome Atlas; LUAD: Lung adenocarcinoma; CAFRG: Cancer-associated fibroblast-related genes; GSEA: Gene set enrichment analysis.

**Figure 8 ijms-26-03478-f008:**
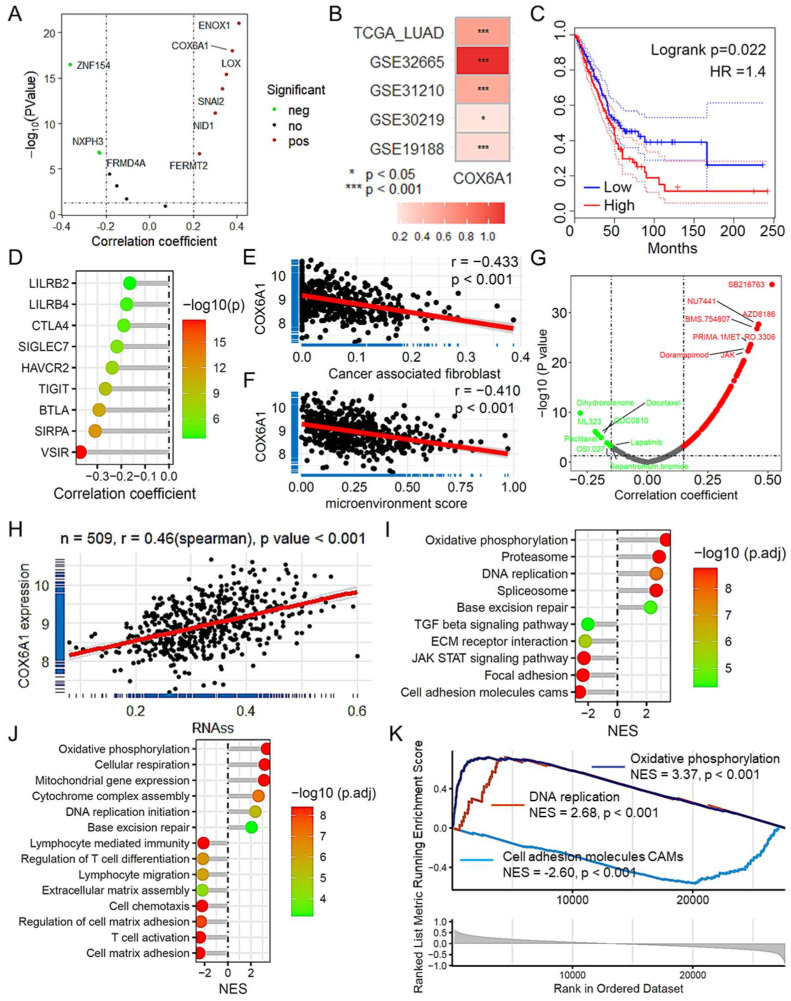
*COX6A1* is a gene that promotes tumor progression in the CAFRG prognostic model. (**A**) The scatter plot illustrates the correlation analysis between CAFRGs and the risk score in the model, revealing a strong association between *COX6A1* and the risk score. (**B**) The heatmap illustrates the expression variation of *COX6A1* in tumor tissues compared with normal tissues across multiple LUAD datasets. (**C**) The survival curve shows the prognostic differences between patients with high and low expression of *COX6A1* in the TCGA LUAD dataset. (**D**) The lollipop plot shows the correlation between *COX6A1* expression and immune checkpoint gene expression in the TCGA LUAD dataset. The scatter plots illustrate the correlation between *COX6A1* expression and cancer associated fibroblast infiltration score (**E**) and microenvironment score (**F**) derived from xCell, highlighting associations with various immune cell types. (**G**) The scatter plot displays the correlation between *COX6A1* expression and antitumor drug sensitivity scores on the basis of OncoPredict, suggesting enhanced drug sensitivity. (**H**) The scatter plot shows the correlation between *COX6A1* expression and tumor stemness assessed by the RNA stemness score (RNAs), indicating a strong positive relationship. (**I**) The lollipop plot depicts signaling pathways associated with *COX6A1* identified by GSEA. (**J**) The lollipop plot reveals biological functions related to *COX6A1* expression. (**K**) The GSEA plot highlights *COX6A1* related to DNA replication, oxidative phosphorylation, and cell adhesion molecules. TCGA: The Cancer Genome Atlas; LUAD: Lung adenocarcinoma; CAFRG: Cancer-associated fibroblast-related gene; GSEA: Gene set enrichment analysis.

**Figure 9 ijms-26-03478-f009:**
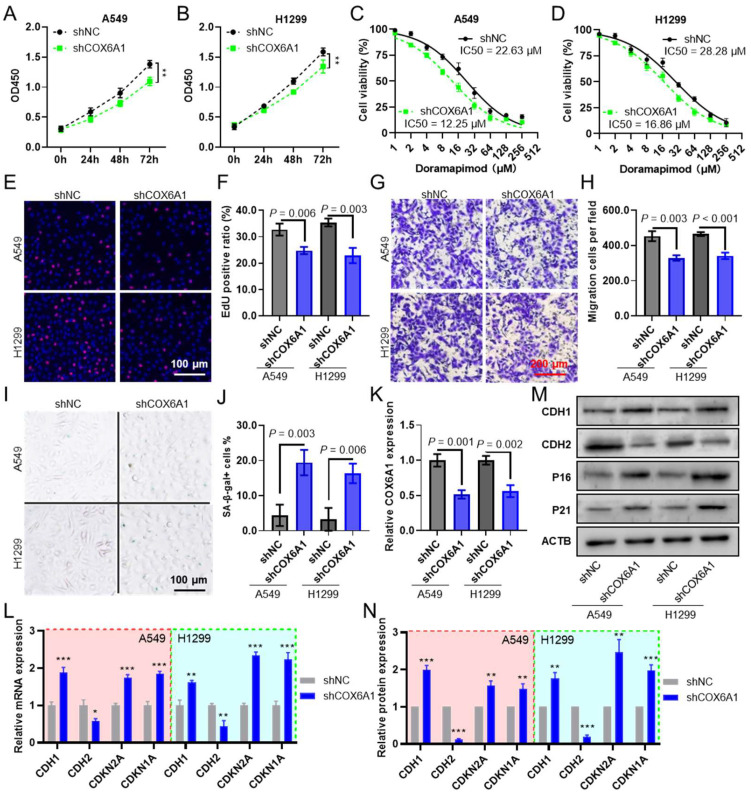
*COX6A1* is a gene that promotes tumor progression in the model. CCK8 assay for analyzing the effects of *COX6A1* knockdown on the proliferation of A549 (**A**) and H1299 (**B**) lung adenocarcinoma cells; CCK8 assay dose–response curves of A549 (**C**) and H1299 (**D**) cells treated with different concentrations of Doramapimod; Transwell migration assay images (**E**) and quantification (**F**) showing the effects of *COX6A1* knockdown on cell migration; EdU proliferation assay images (**G**) and quantification (**H**) showing the impact of *COX6A1* knockdown on cell proliferation; β-galactosidase staining images (**I**) and quantification (**J**) showing the induction of cellular senescence following *COX6A1* knockdown; qPCR analysis showing changes in *COX6A1* (**K**) and migration and senescence-related genes (**L**) after knockdown; Western blot analysis images (**M**) and quantification (**N**) demonstrating changes in migration and senescence-related proteins. This section may be divided into subheadings. It should provide a concise and precise description of the experimental results, their interpretation, as well as the experimental conclusions that can be drawn. Compare with shNC, * *p* < 0.05, ** *p* < 0.01, *** *p* < 0.001.

**Figure 10 ijms-26-03478-f010:**
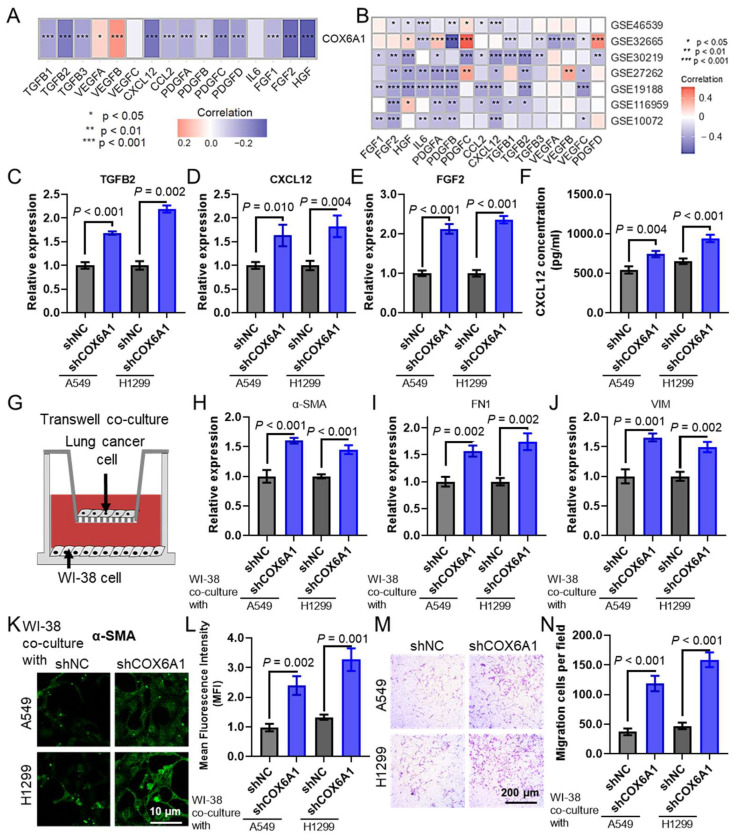
*COX6A1* overexpression in lung cancer cells promotes CAF infiltration. Heat maps show the correlation between *COX6A1* and the expression of CAF activation-related cytokines in the TCGA LUAD dataset (**A**) and multiple LUAD datasets from the GEO database (**B**). qPCR analysis of *COX6A1*-knockdown lung cancer cells shows changes in the expression of *TGFB2* (**C**), *CXCL12* (**D**), and *FGF2* (**E**). (**F**) ELISA analysis of CXCL12 levels in the culture supernatant of *COX6A1*-knockdown lung cancer cells. (**G**) A co-culture system of lung cancer cells and human embryonic lung cells WI-38. qPCR analysis of RNA expression of α-*SMA* (**H**), *FN1* (**I**), and *VIM* (**J**) in WI-38 cells co-cultured with lung cancer cells. Immunofluorescence images (**K**) and quantitative results (**L**) showing α-SMA expression in WI-38 cells co-cultured with lung cancer cells. Transwell migration assay images (**M**) and quantitative results (**N**) show the migratory capacity of WI-38 cells co-cultured with lung cancer cells. TCGA, The Cancer Genome Atlas; LUAD, Lung Adenocarcinoma; GEO, Gene Expression Omnibus.

**Table 1 ijms-26-03478-t001:** Demographic and clinical characteristics of the training, internal testing, and complete TCGA LUAD datasets.

Characteristics		TCGA LUAD			Chi-Square *p* Value
		Training (*n* = 353)	Internal Testing (*n* = 151)	All (*n* = 504)	
Gender	female	194 (54.96%)	76 (50.33%)	270 (53.57%)	0.634
	male	159 (45.04%)	75 (49.67%)	234 (46.43%)	
Age	≤60	119 (34.59%)	39 (26.00%)	158 (31.98%)	0.170
	>60	225 (65.41%)	111 (74.00%)	336 (68.02%)	
M	M0	231 (90.59%)	100 (95.24%)	335 (93.06%)	0.116
	M1	24 (9.41%)	5 (4.76%)	25 (6.94%)	
N	N0	225 (65.79%)	99 (66.89%)	324 (66.12%)	0.972
	N1/2	117 (34.21%)	49 (33.11%)	166 (33.88%)	
T	T1/2	306 (86.69%)	132 (87.42%)	438 (86.90%)	0.975
	T3/4	47 (13.31%)	19 (12.58%)	66 (13.10%)	
Stage	Stage I/II	272 (77.05%)	118 (78.15%)	390 (77.38%)	0.965
	Stage III/IV	81 (22.95%)	33 (21.85%)	114 (22.62%)	
Smoke history	Nonsmoke	139 (39.38%)	61 (40.40%)	200 (39.68%)	0.977
	Smoke	214 (60.62%)	90 (59.60%)	304 (60.32%)	
Time	≤2	204 (57.79%)	81 (53.64%)	285 (56.55%)	0.691
	>2	149 (42.21%)	70 (46.36%)	219 (43.45%)	
Status	0	220 (62.32%)	101 (66.89%)	321 (63.69%)	0.621
	1	133 (37.68%)	50 (33.11%)	183 (36.31%)	

Note: TCGA: The Cancer Genome Atlas; LUAD: Lung adenocarcinoma.

## Data Availability

All the data generated and described in this article are available from the corresponding web servers and are freely available to any scientist wishing to use them for non-commercial purposes without breaching participant confidentiality. All codes and R packages used in the study are publicly available and have been disclosed in Section 4 or are available from the corresponding authors upon reasonable request.

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
