# Peer review of "Development and Validation of a Prognostic Model for Lung Adenocarcinoma Based on CAF-Related Genes: Unveiling the Role of COX6A1 in Cancer Progression and CAF Infiltration"

_ijms, 2025, doi:10.3390/ijms26083478_

Round 1

Reviewer 1 Report

Comments and Suggestions for Authors

In this study, Zhu et al., sought to create a prognostic model for patients with lung adenocarcinoma based on genes linked to tumor-associated fibrobast infiltration scores, while also confirming its accuracy and reliability. The main goals involved identifying CAF-related genes, developing a prognostic model, validating it across independent cohorts, evaluating the clinical importance of risk scores, and performing in vitro functional validation of crucial genes. The manuscript is well written and cover an interesting topic in oncobiology. I have some suggestions and hope to contribute to improve the quality of the paper.

To facilitate the review, the authors should number the lines of the manuscript.

In the introduction section, the authors should write in more detail about the impact of CAFs on events associated with cancer progression, particularly lung cancer. This will help readers who are interested in the topic but do not work directly in the field to better understand it.

It has been well described elsewhere, that The COX6A1 gene encodes a protein that is part of the cytochrome c oxidase (COX) complex, which plays a critical role in the mitochondrial respiratory chain. The authors demonstrate that COX6A1 emerged as a key gene in the model, with its upregulation associated with immune cell infiltration and immune escape. However, they mention little about what is already described in the literature regarding COX6A1 in events associated with the progression of different types of cancer. I suggest that the authors revisit, especially the discussion section of the paper, and cite relevant works in the field that were unmentioned throughout the manuscript.

Why did the authors choose the A549 and H1299 cell lines for the in vitro assays? It is already known that A549 is a heterogeneous cell line that retains epithelial characteristics, while H1299 is a p53-null cell line. Could the authors explain the choice? 

Regarding drug sensitivity, the two cell lines show some important molecular differences. The authors conducted some drug sensitivity assay with both A549 and H1299 cell lines?

I suggest that the authors add the original reference at the end of each methodology used, even if the methods employed in the study have already been published by the same group.

I suggest that the authors add to the manuscript the manufacturer of the antibodies used, as well as the dilution used in the biological assays. This type of information is important.

Please provide the sequences of the primers used in the study. The sequences are not available in the main text and are also not available in the supplementary material.

The authors provided the original Western blot images, but they are cropped (the entire membrane is not shown). Did the authors crop the same membrane to detect different proteins?

I thank the authors for providing a table with the abbreviations. This greatly facilitated the reading of the manuscript.

In the statistical analysis, the authors mention that differences between two groups were assessed via Student’s t-test, whereas comparisons among multiple groups were made via analysis of variance (ANOVA). Regarding comparisons among multiple groups, did the authors use a post-test? 

Author Response

Comments 1: To facilitate the review, the authors should number the lines of the manuscript.

Response 1: Thank you for the suggestion. We are using the template provided by the journal for editing. We believe it would be beneficial to suggest to the journal that line numbers be added to manuscripts intended for review.

Comments 2: In the introduction section, the authors should write in more detail about the impact of CAFs on events associated with cancer progression, particularly lung cancer. This will help readers who are interested in the topic but do not work directly in the field to better understand it.

Response 2: Thank you for the suggestion. We have added more detailed information about the impact of cancer-associated fibroblasts (CAFs) on cancer progression, particularly lung cancer, in the introduction section to help readers better understand the topic.

Comments 3: It has been well described elsewhere, that The COX6A1 gene encodes a protein that is part of the cytochrome c oxidase (COX) complex, which plays a critical role in the mitochondrial respiratory chain. The authors demonstrate that COX6A1 emerged as a key gene in the model, with its upregulation associated with immune cell infiltration and immune escape. However, they mention little about what is already described in the literature regarding COX6A1 in events associated with the progression of different types of cancer. I suggest that the authors revisit, especially the discussion section of the paper, and cite relevant works in the field that were unmentioned throughout the manuscript.

Response 3: Thank you for your suggestion. In fact, we conducted an extensive literature search during the writing of our paper and did not find relevant research reports on COX6A1 in cancer. We recognize the potential value of this gene in cancer research and are currently conducting a series of in vitro experiments to study its function.

Comments 4: Why did the authors choose the A549 and H1299 cell lines for the in vitro assays? It is already known that A549 is a heterogeneous cell line that retains epithelial characteristics, while H1299 is a p53-null cell line. Could the authors explain the choice? 

Response 4: In our study, we selected the A549 and H1299 lung cancer cell lines for in vitro assays for several reasons. Firstly, A549 is a well-characterized adenocarcinoma cell line with a KRAS mutation, which makes it a valuable model for studying this common molecular subtype in lung cancer. Secondly, H1299 is a p53-null cell line derived from lymph node metastatic non-small cell lung cancer (NSCLC), representing another prevalent genetic background in lung cancer.

Using these two widely-used classic lung cancer cell models allows us to cover a broader spectrum of lung cancer biology, thereby enhancing the generalizability and relevance of our findings. By incorporating both cell lines, we aim to avoid the limitations associated with using a single cell line and ensure that our results are robust and applicable to different molecular subtypes and genetic backgrounds of lung cancer.

We appreciate the reviewer's insightful comments and hope this explanation clarifies our rationale for selecting these cell lines.

Comments 5: Regarding drug sensitivity, the two cell lines show some important molecular differences. The authors conducted some drug sensitivity assay with both A549 and H1299 cell lines?

Response 5: Thank you for your question. In this study, we only explored the sensitivity of the drug doramapimod, identified through bioinformatics analysis, in the two cell lines. Other drugs were not tested. We will consider the differences between the two cell lines in our future in-depth studies.

Comments 6: I suggest that the authors add the original reference at the end of each methodology used, even if the methods employed in the study have already been published by the same group.

Response 6: We appreciate the reviewer's suggestion to add the original references at the end of each methodology used in our study. We understand the importance of comprehensive citation for reproducibility and acknowledgment of prior work. In our manuscript, we have included references for key analysis packages and methods that are critical to our study.

For commonly used and well-established methods, we have not provided individual references as these techniques are widely recognized and standard in the field. We believe that including references for all methodologies, especially for routine procedures, may not significantly enhance the clarity or reproducibility of our study. However, we are willing to review the manuscript and ensure that all critical and specialized methods are appropriately referenced. We hope this approach strikes a balance between thorough citation and maintaining the manuscript's readability.

Thank you for your valuable suggestion, which helps us improve the quality of our manuscript.

Comments 7: I suggest that the authors add to the manuscript the manufacturer of the antibodies used, as well as the dilution used in the biological assays. This type of information is important.

Response 7: Thank you for your valuable suggestions. We fully understand the importance of providing information about the antibody sources and dilution factors in enhancing the transparency and reproducibility of the experiments. Based on your advice, we have added detailed information about the antibodies used in the manuscript, including the manufacturer, dilution factors, and catalog numbers for each antibody.

Comments 8: Please provide the sequences of the primers used in the study. The sequences are not available in the main text and are also not available in the supplementary material.

Response 8: Thank you for your valuable suggestions. We fully understand the importance of providing primer sequences in ensuring the reproducibility and scientific validity of the experimental results. Based on your advice, we have added the sequences of all primers used in the manuscript text.

Comments 9: The authors provided the original Western blot images, but they are cropped (the entire membrane is not shown). Did the authors crop the same membrane to detect different proteins?

Response 9: Thank you for your valuable feedback. As you mentioned, we indeed cropped the same membrane to detect multiple proteins. We recognize the importance of showing the full membranes and incubating antibodies as well as developing them to ensure the transparency of the experiments and the integrity of the data.

In future studies, we will pay special attention to this and ensure that full membrane images are provided to further enhance the transparency and reproducibility of our research.

Comments 10: I thank the authors for providing a table with the abbreviations. This greatly facilitated the reading of the manuscript.

Response 10: We appreciate your acknowledgment and will continue to strive for clarity and accessibility in our work.

Comments 11: In the statistical analysis, the authors mention that differences between two groups were assessed via Student’s t-test, whereas comparisons among multiple groups were made via analysis of variance (ANOVA). Regarding comparisons among multiple groups, did the authors use a post-test? 

Response 11: Thank you for your thoughtful question regarding the statistical analysis in our manuscript. We would like to clarify that our study did not involve comparisons among multiple groups, and therefore, ANOVA was not applicable. In light of this, we have removed the relevant description from the statistical analysis section of the manuscript.

Reviewer 2 Report

Comments and Suggestions for Authors

The authors presented a model not only effective to predict the prognosis of LUAD patients but also anticipate with the tumor immune microenvironment, drug sensitivity, and tumor biological characteristics, such as proliferation, migration. I agree with the choice of gene sequencing and the results are quite expected and interesting. The presentation of the survival curves are interesting too. Therefore, I suggest minor revision (noted).

The authors did not discuss the impact of ECM remodelling in this paper. However, extracellular matrix (ECM) remodeling and cancer-associated fibroblasts play a crucial role in the regulation of cancer aggressiveness and there is a growing need to investigate their role in the determination of LUAD behavior at early stages. Authors should correlate the possible antagonist/agonistic impact of the model in that direction.

The authors must specify the pathways on which their model could enlighten deeper unmderstanding like angionesis, or metastasis or resistance, etc.?   

Needs to add more correlative details in the experimental part and analysis portion to add model design support.

The authors did not cite very closely relevant diagnostic reports from recent papers like Sci Rep 13, 17604 (2023); Cancers 202416(24), 4180, etc.

Author Response

Comments 1: The authors did not discuss the impact of ECM remodelling in this paper. However, extracellular matrix (ECM) remodeling and cancer-associated fibroblasts play a crucial role in the regulation of cancer aggressiveness and there is a growing need to investigate their role in the determination of LUAD behavior at early stages. Authors should correlate the possible antagonist/agonistic impact of the model in that direction.

Response 1: Thank you for pointing this out. We included a discussion on ECM remodeling and its role in early LUAD behavior in the revised manuscript.

Comments 2: The authors must specify the pathways on which their model could enlighten deeper unmderstanding like angionesis, or metastasis or resistance, etc.?   

Response 2: Thank you for the suggestion. We specified in the revised manuscript the key pathways our model may shed light on, such as angiogenesis, metastasis, and resistance, to further deepen the understanding of LUAD biological behavior.

Comments 3: Needs to add more correlative details in the experimental part and analysis portion to add model design support.

Response 3: Thank you for your input. We will incorporate additional experimental and analytical details related to the model design to support our findings.

Comments 4: The authors did not cite very closely relevant diagnostic reports from recent papers like Sci Rep 13, 17604 (2023); Cancers 202416(24), 4180, etc.

Response 4: Thank you for the recommendation. We will include citations of recent relevant diagnostic reports, such as Sci Rep 13, 17604 (2023) and Cancers 2024, 16(24), 4180, to ensure comprehensiveness and currency of the literature review.

Round 2

Reviewer 1 Report

Comments and Suggestions for Authors

I thank the authors for improving the manuscript.